# Removal of Iron(II) from Effluents of Steel Mills Using Chemically Modified *Pteris vittata* Plant Leaves Utilizing the Idea of Phytoremediation

Qaiser Khan [1], Muhammad Zahoor [2,*], Syed Muhammad Salman [1], Muhammad Wahab [1], Farhat Ali Khan [3], Naila Gulfam [4] and Ivar Zekker [5,*]

[1] Department of Chemistry, Islamia College University, Peshawar 25000, Pakistan; qaisaricu@gmail.com (Q.K.); salman@icp.edu.pk (S.M.S.); mwahabbajaur@gmail.com (M.W.)

[2] Department of Chemistry, University of Malakand, Dir Lower 18800, Pakistan

[3] Department of Pharmacy, Shaheed Benazir Bhutto University, Dir Upper 18000, Pakistan; farhatkhan2k9@yahoo.com

[4] Department of Zoology, Jinnah College for Women, University of Peshawar, Peshawar 23000, Pakistan; nailazoo@yahoo.com

[5] Institute of Chemistry, Faculty of Science, University of Tartu, 14 Ravila St., 50411 Tartu, Estonia

[*] Correspondence: mohammadzahoorus@yahoo.com (M.Z.); ivar.zekker@ut.ee (I.Z.)

**Abstract:** Dargai District Malakand, Pakistan, is a tax-free zone that attracts many industrialists to install their plants in this area. Along with other industries, a number of steel mills are polluting the natural environment of this locality. This study aimed to evaluate heavy metals levels in steel mills effluents and fabricate an efficient adsorbent from the leaves of plants growing on the banks of the drainage lines of the industries and having high phytoremediation capabilities, through chemical modifications. Initially, the effluents were analyzed for heavy metal concentrations, then the leaves of a plant (*Pteris vittata*) with better phytoremediation capability were chemically modified. The leaves of *Pteris vittata* were crushed into a fine powder, followed by chemical modification with $HNO_3$, then washed with distilled water, neutralized with NaOH and finally activated through calcium chloride to enhance its biosorption ability, abbreviated as CMPVL. Fourier transform infrared spectroscopy (FTIR), scanning electron microscopy (SEM), surface area analyzer, energy dispersive X-ray spectroscopy (EDX), and thermal gravimetric analysis (TGA) were used to characterize the CMPVL. The modified leaves in the powdered form were then used for the reclamation of Fe(II) present in the effluents of the mentioned industries. Batch biosorption tests were performed under varied physicochemical conditions of pH (2–9), contact time (10–140 min), temperature (293–333 K), biosorbent dose (0.01–0.13 g), and initial metal concentration (20–300 mg $L^{-1}$) to optimize the removal of the selected metal. Langmuir, Jovanovic, Freundlich, Temkin, and Harkins–Jura isotherm models were used to assess the equilibrium data. With a high $R^2$ value of 0.977, the Langmuir model offered an excellent match to the equilibrium data. The pseudo-first order, pseudo-second order, power function, intraparticle diffusion, and Natarajan–Khalaf models were applied to experimental kinetics data. With $R^2$ values of 0.999, the pseudo-second order model well fitted the obtained data. The Van't Hoff equation was used to calculate $\Delta H°$, $\Delta S°$ and $\Delta G°$ of Fe(II) sorption on CMPVL. The $\Delta H°$ and $\Delta G°$ were negative, whereas $\Delta S°$ was positive, suggesting that the biosorption process was exothermic, favorable, and spontaneous. The selected plant leaves were found to be efficient in the reclamation of iron from the industrial effluents (as the plant has a high natural capability for remediating the selected metal ion) after chemical modification and may be used as an alternative to activated carbon as being a low-cost material and a high phytoremediator of iron metal. Such natural phenomena of phytoremediation should be utilized in obtaining efficient adsorbents for other metals as well.

**Keywords:** iron; adsorption; phytoremediation; industrial effluents

## 1. Introduction

Although some heavy metals are required in minute concentrations for human and plant growth, their overabundance in the human diet or water is harmful. As a result of global industrialization, water, an essential component of life, is susceptible to heavy metal contamination. To obtain safe drinking water, the removal of heavy metals and other pollutants from water is thus mandatory. The presence of heavy metals in water has the potential to harm aquatic species as well as humans. Heavy metals penetrate the food chain and have a negative impact on human health [1–4]. These metals persist for a long period in the aquatic environment and in soil because of their non-biodegradable nature. As a result, various methods have been tested to remove them from industrial effluents before reaching freshwater bodies [5]. Iron being the essential metal for infrastructure development is constantly added to water bodies and it is mandatory that its level must be kept below the permissible concentration limits, otherwise it will adversely affect the lives of humans and plants. Automobile, aviation, paint, and steel mills all produce enormous amounts of effluent with varying levels of iron [6]. Furthermore, water moving over rocks and soil also dissolves minerals, such as iron into the water stream, thereby raising their concentrations [7]. When the iron content in the water exceeds 0.3 mg $L^{-1}$ (the permissible level), water staining occurs, causing damage to drainage pipes, dinnerware, and textiles, as well as causing a yellow to reddish coloration in the water itself. The taste and odor of drinking water may be affected by high iron content [8].

Heavy metals can be removed from aqueous media using a variety of processes including, electrolysis, flocculation, photocatalytic degradation, ion exchange process, oxidation, and even membrane technologies [9–18]. Regrettably, the in-use treatment systems have a number of disadvantages, such as high operating and maintenance costs, complex procedures, high chemical utilization, and processing that generates a lot of secondary sludge which in some cases may be hazardous [19–21]. Furthermore, at low heavy metal concentrations, less than 100 mg $L^{-1}$, the traditional treatment methods are inefficient and ineffectual [22,23]. These days biological remedies are constantly investigated to resolve environmental issues and be compatible with nature. Phytoremediation is a biological phenomenon where plants can accumulate heavy metals and other hazardous substances from the soil. For example, *Pteris vittata* is capable of hyperaccumulating arsenic from the environment. However, phytoremediation practices are of limited use as most plants are edible, also plant growth is very slow, making it impractical [24]. Adsorption is the most reliable and practical method for removing metal ions from the aquatic environment. The biosorption technique is a widespread technique used for the restoration of the aquatic environment because of its minimal price, environment friendly nature, and use of natively and widely accessible biomass in the fabrication of adsorbents [25]. At the same time, it is quite easy to regenerate biomass-based adsorbents, which are among the most essential elements for determining the cost of a given water treatment technique [26–28]. The natural phenomenon of phytoremediation has not been utilized by the scientific community in the fabrication of a biosorbent. The waste biomass of a plant having high phytoremediation capacity for a given metal ion would most probably have a high adsorption capacity as well if converted into an adsorbent.

In the present study, a novel approach has been used in the fabrication of an efficient adsorbent, where the idea of phytoremediation has been combined with the widely used idea of adsorption to fabricate the biosorbent. The prepared biosorbent was then used for the elimination of iron from the industrial effluents of steel mills located in Dargai, District Malakand, Pakistan. Initially, plants growing at the banks of industrial drainage lines were screened out for the phytoremediation capability of iron ions and then among the tested plants, the best phytoremediator's leaves were converted into adsorbents. Such ideas have not been utilized in the literature by any researchers yet. As per our unpublished results, *Pteris vittata* (also known as Chinese brake), a common plant found in the locality where steel mills are installed was found to remediate iron ions effectively. Its leaves were modified chemically for use as an efficient adsorbent for the iron present in the effluents

(as mentioned before, plant growth is a slow process, therefore, phytoremediation is less effective in comparison if used as an adsorbent). Various instrumental techniques including FTIR, SEM, EDX, TG/DTA, and surface area analyzer were utilized to characterize the adsorbent. Batch tests were performed to optimize the biosorption process. The stability of an adsorbent is a critical parameter, particularly when the same adsorbent is employed in many adsorption cycles. As a result, the adsorbent's reusability was also assessed.

## 2. Material and Methods

### 2.1. Selection of the Plant

Initially, the steel mills' effluents were screened out for heavy metal concentrations where iron was found to be present in high concentrations. Then a number of plants were screened out for their phytoremediation capabilities. Out of the tested plants *Pteris vittata* was selected to be converted into a biosorbent due to its high phytoremediation capabilities (our unpublished results). The leaves of the selected plants were collected from the Dargai Malakand district of KPK, Pakistan where the steel mills are abundantly installed. They were rinsed with tap water first, then twice with distilled water to remove dust particles and soluble contaminants. Then placed in the shade until complete dryness. The crisped leaves were placed in an electrical oven set at 40 °C for 24 h to remove the remaining moisture content. The leaves were then crushed and sieved to 44-mesh size particles using an electric grinder.

### 2.2. Chemical Modification

Approximately 50 g of powdered sample was soaked in a 1 L $HNO_3$ (0.1 M) solution for 24 h then filtered through 42 Whatmann filter paper and washed many times with distilled water. For neutralization, 0.1 M NaOH was employed. The neutralized biosorbent was dried at ambient temperature before oven drying at 100 °C. After that, about 50 g of it was activated with 1 L $CaCl_2$ solution (0.1 M). Following activation, the biosorbent was dried in an oven. The $HNO_3$ solution was used to extract already bound metals from the biomass feedstock, while the $CaCl_2$ treatment was used to add a definite group to the biosorbent, which would aid in Fe ion exchange from the solution and adsorbent.

### 2.3. Characterization of Biosorbent

For the estimation of surface area and pore volume, the ISO-8962 method using a pycnometer was used. The FTIR spectra (Thermo Fisher Nicoletis 50; Madison, WI, USA) of chemically modified Pteris vittata leaves (CMPVL) before and after biosorption were obtained in the frequency range 400–4000 $cm^{-1}$ using the KBr pellet method. The morphology of treated and untreated Pteris vittata leaves was studied using SEM while the stability of the biosorbent was determined through TGA.

### 2.4. Adsorption Experiments

To obtain working standards (20 to 300 mg $L^{-1}$), a stock solution of $FeSO_4 \cdot 7H_2O$ was made and then diluted with distilled water. In a batch experiment, 50 mL of $FeSO_4 \cdot 7H_2O$ working standards were mixed with 0.09 g of biomass and agitated for 2 h at 120 rpm. Whatmann Filter Paper was used to filter the solutions. The Fe(II) ion contents in the filtrate were measured through atomic absorption spectrophotometer. Equations (1) and (2) were used to compute the biosorption capacity $q_e$ (mg/g) and Fe(II) percent uptake by the biosorbent from the solution.

$$q_e = (C_i - C_f) \times \frac{V}{m} \tag{1}$$

$$\% \, \text{R} = \frac{(C_i - C_f)}{C_i} \times 100 \tag{2}$$

where $C_i$ represents the initial Fe(II) concentration and $C_f$ shows the final Fe(II) concentration. The volume of the solution is expressed in liters, while the quantity of the biosorbent is expressed in grams.

### 2.5. Isotherm Study

The biosorption of Fe(II) on CMPVL was tested by increasing the starting Fe(II) concentration from 20 to 300 mg L$^{-1}$. The experimental conditions were the pH of the solution = 6, the volume of the solution = 50 mL, the contact time = 120 min, and the mass of the biosorbent = 0.09 g. Equation (1) was used to obtain the $q_e$ values. The biosorption equilibrium data were evaluated using Freundlich, Langmuir, Temkin, Jovanovic, and Harkin–Jura models.

### 2.6. Adsorption Kinetics

To evaluate the kinetic parameters of Fe(II) biosorption on CMPVL, a fixed amount of the biosorbent (0.09 g) was added to 50 mL of Fe(II) solution (50 mg L$^{-1}$). The solutions were stirred at 120 rpm for 2 h. The kinetics data of biosorption were fitted into pseudo-first-order, pseudo-second-order, intraparticle diffusion models, Power function, and Natarajan and Khalaf models.

### 2.7. Effect of pH and Biosorbent Mass

The role of pH on Fe(II) biosorption was studied in the range from 2 to 9. The pH of the solution was adjusted through $HNO_3$ (0.1 M) and NaOH (0.1 M). From 0.01 to 0.13 g L$^{-1}$, the impact of the biosorbent mass was examined as well.

### 2.8. Evaluation of Thermodynamic Parameters and Regeneration of the Biosorbent

Temperature influences on Fe(II) biosorption onto CMPVL were evaluated at 293 K, 303 K, 313 K, 323 K, and 333 K keeping other parameters constant (mentioned above). The biosorbent was reused for five cycles after regeneration with dilute HCl and deionized water. The adsorption capacity was determined after each cycle and the percent efficiency was calculated.

## 3. Result and Discussion

### 3.1. Characterization of CMPVL

3.1.1. FTIR Spectra of Unloaded and Loaded Fe(II) CMPVL

The unloaded and loaded CMPVL FTIR spectrum is shown in Figure 1A,B. The peak at 3200 cm$^{-1}$ to 3400 cm$^{-1}$ indicates the N-H group. The peak at 3000–3100 cm$^{-1}$ is due to the C-H stretch. Similarly, the peak between 1630 and 1680 cm$^{-1}$ is due to carbonyl group stretching. The peak at 600 cm$^{-1}$ is due to the N–H stretching whereas the peak at 1100–1300 cm$^{-1}$ represents C-N stretching. The loaded adsorbent represents drastic changes from the unloaded one indicating the interactions of Fe(II) with the functional groups present on CMPVL.

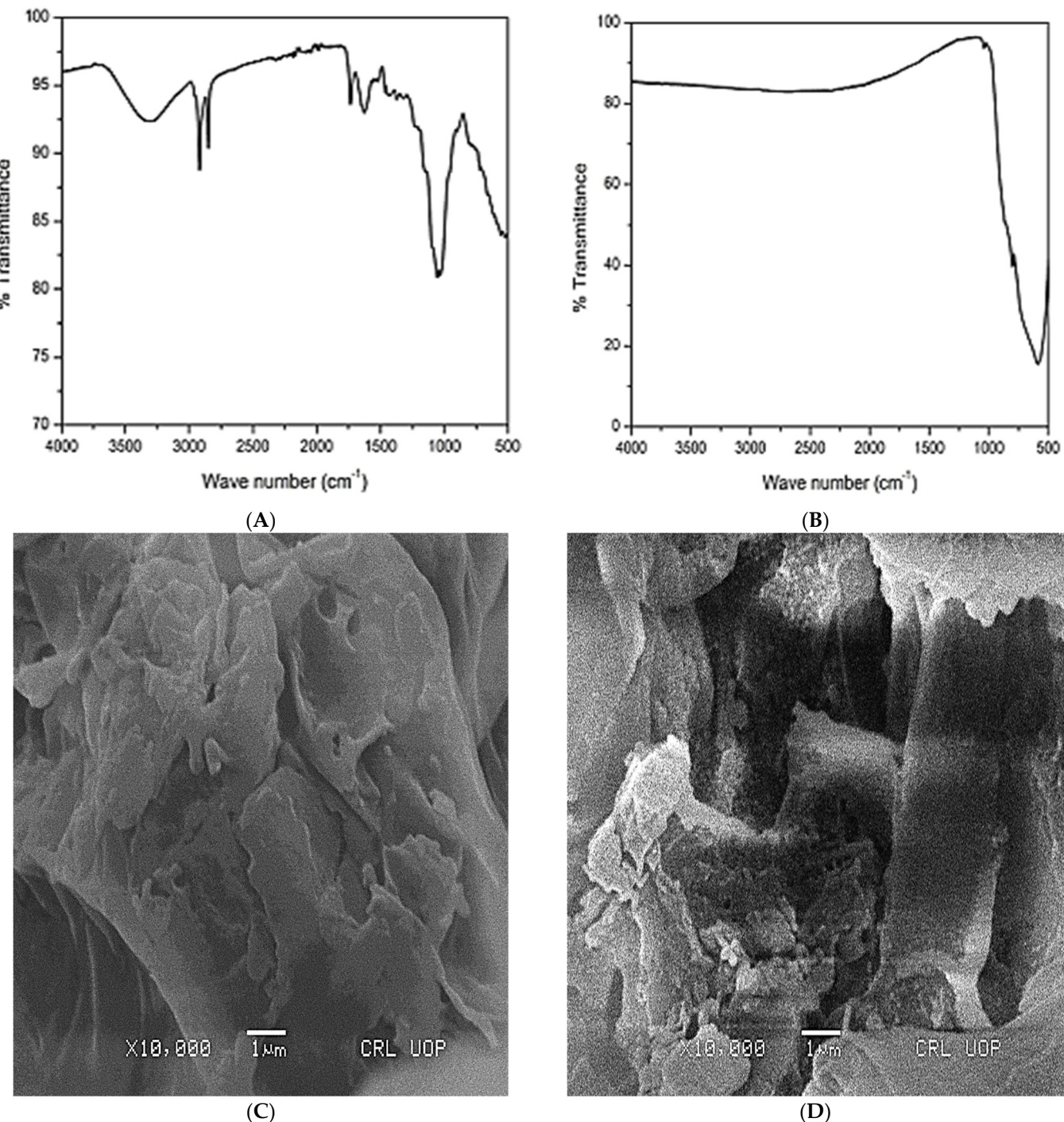

**Figure 1.** *Cont*.

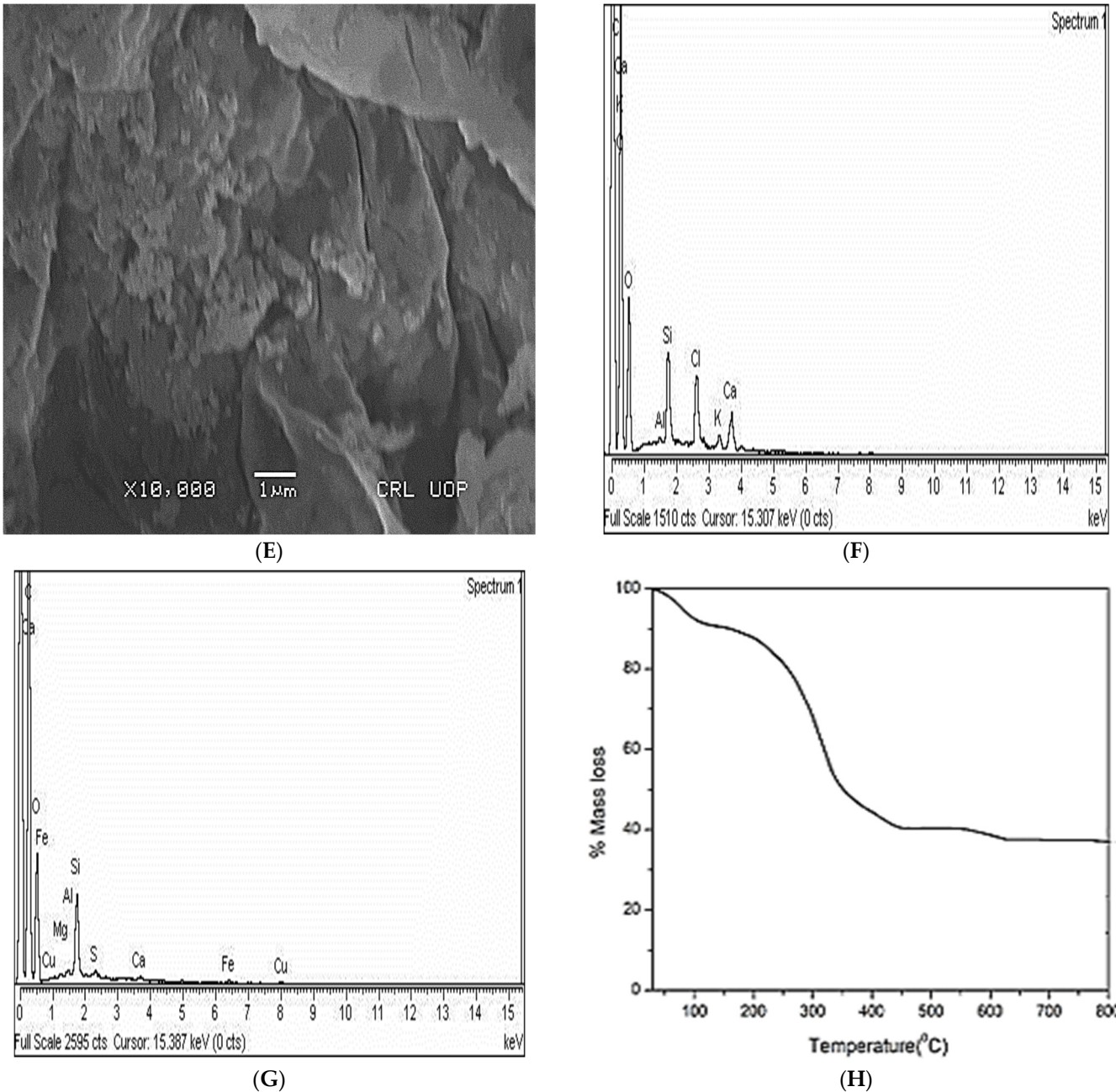

**Figure 1.** *Cont.*

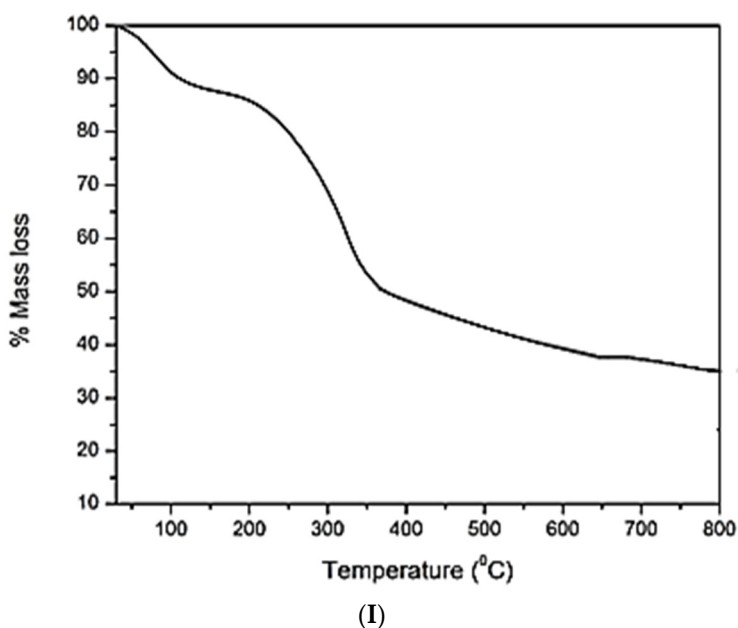

**(I)**

**Figure 1.** *Cont*.

3.1.2. Surface Area, and Pore Volume

The surface area and pore volume are given in Table 1. According to the table, CMPVL has a larger surface area and better pores distribution, making it a preferable biosorbent in terms of biosorption capacity determined through the use of different isotherms.

**Table 1.** Surface area, pore volume and diameter of CMPVL.

| Parameter | Value |
| --- | --- |
| BET surface area ($m^2/g$) | 73.28 |
| Pore volume (cc/g) | 0.82 |

3.1.3. SEM Images of Treated and Untreated Biosorbent

Figure 1C–E illustrates SEM images of untreated, treated, and iron-loaded biosorbent. The SEM images showed multi porous faces on the treated biosorbent surface having indefinite shapes with shattered edges and crooks, showing its capabilities to be used as an efficient biosorbent. Comparing the images, important structural dissimilarity can be observed among them. The uniformity in the structure of the treated biosorbent is evident as compared to untreated powders.

3.1.4. EDX of Unloaded and Loaded Biosorbent

The elemental analysis (EDX) of treated and iron loaded biosorbent is shown in Figure 1F,G. When compared to other minor peaks of Mg, Si, P, S, Cl, K, and Ca that occurred as impurities, the carbon and oxygen peaks are more apparent. The iron loaded biosorbent showed distinct peaks of iron indicating the biosorption of iron on the prepared biosorbent.

3.1.5. Thermal Gravimetric Analysis

The TGA spectra of untreated and treated biosorbent are shown in Figure 1H,I. The mass of the untreated biosorbent analyzed was 8.286 mg. The mass loss from 0 to 100 °C was 9%, which was attributed to water evaporation. The second mass loss was seen up to 450 °C and was attributed to the heat breakdown of the cellulose. The production of carbonaceous residues was considered to be the third mass loss recorded above 640 °C. The treated biosorbent mass taken for analysis was 7.873 mg, where the first mass loss was observed at 150 °C which may be due to the loss of water molecules. The formation of

carbonaceous materials is caused by the heat degradation of cellulose, which causes the second mass loss at 650 °C and then the mass remained stable at around 750 °C.

### 3.2. Adsorption Isotherms

According to existing research, the removal of metal by a specific adsorbent increases with a rise in initial concentration within certain limits and beyond that, there will be no further increase in adsorption capacity once the adsorbent reaches its saturation point [29]. Figure 2A depicts the impact of initial iron(II) concentration on biosorption onto the biosorbent. With increasing the initial iron(II) concentration, the adsorption efficiency of CMPVL increases as more vacant sites on the adsorbent were available; however, once most of the pores were engaged, a decline in the biosorption can be noticed (clear from the plateau of the curve). The equilibrium data were fitted into Freundlich, Langmuir, Temkin, Jovanovic, and Harkins–Jura models as per the following details.

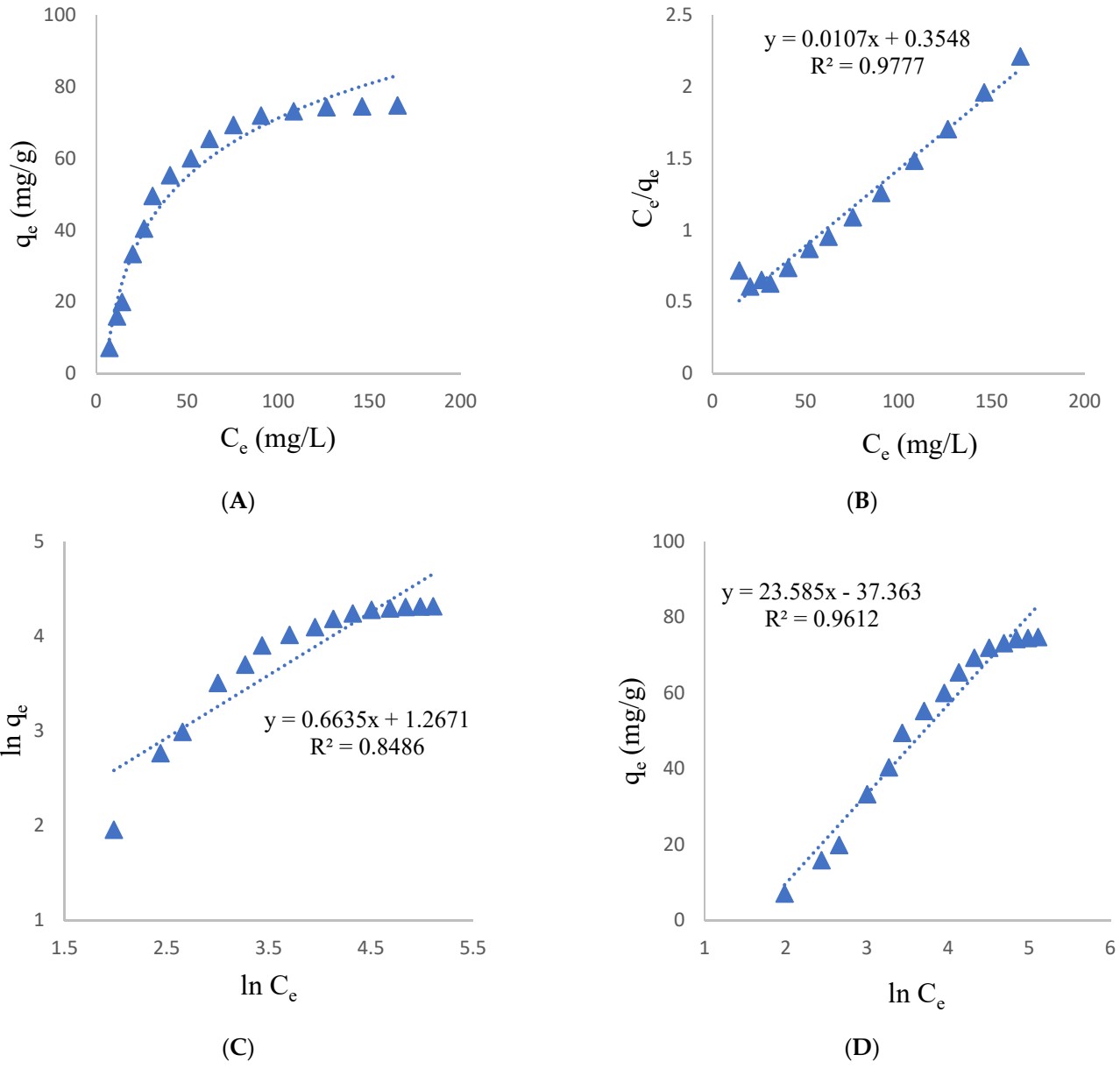

**Figure 2.** *Cont.*

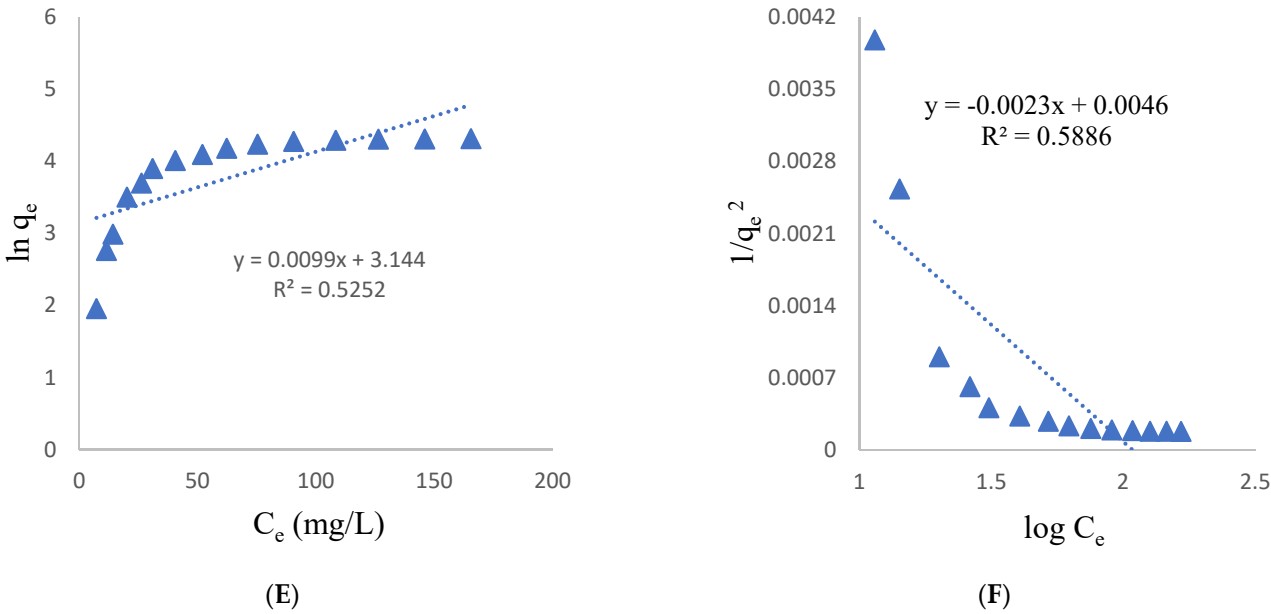

**Figure 2.** Isotherm applied: (**A**) Impact of initial Fe(II) concentration (**B**) Langmuir isotherm plot (**C**) Freundlich plot (**D**) Temkin isotherm plot (**E**) Jovanovic isotherm plot (**F**) Harkins jura isotherm plot.

### 3.2.1. Langmuir Isothermal Model

The linear version of the Langmuir model [30] can be given as:

$$\frac{C_e}{q_e} = \frac{1}{q_{max \times b}} + \frac{C_e}{q_e} \tag{3}$$

The $q_e$ denotes the quantity of Fe(II) adsorbed per unit mass of CMPVL at equilibrium, $q_{max}$ represents the highest adsorption capacity, and $C_e$ represents the Fe(II) equilibrium concentration. The term b refers to the Langmuir constant, which is related to binding strength and free energy. Figure 2B depicts a linearized graph of the Langmuir model plotted as $C_e$ versus $C_e/q_e$ where $q_{max}$ and b are the slope and intercept of the plotted data. The $R^2$ value for this model is 0.977 indicating the compatibility of the obtained data with this model.

### 3.2.2. Freundlich Isothermal Model

Adsorption in multilayers with different energy sites can be explained using this model [31]. The following is the linearized form of this model:

$$\ln q_e = \log KF + \frac{1}{n}\ln C_e \tag{4}$$

where $q_e$ is the quantity of Fe(II) adsorbed per unit mass of CMPVL (mg g$^{-1}$), $C_e$ represents the Fe(II) concentration in equilibrium (mg L$^{-1}$), $K_F$ is the Freundlich constant, and n is the adsorption coefficient. Figure 2C shows the slope and intercept of the $\ln q_e$ vs. $\ln C_e$ plot, and Table 2 shows their values. The $R^2$ value for the Freundlich adsorption isotherm was 0.848 which is less than that recorded for the Langmuir model.

**Table 2.** Isothermal parameters for adsorption of Iron(II) on CMPVL.

| Isotherm | Parameters | Values |
|---|---|---|
| Langmuir | | |
| | $q_{max}$ (mg g$^{-1}$) | 100 |
| | $K_L$ (Lmg$^{-1}$) | 0.0282 |
| | $R^2$ | 0.977 |
| Freundlich | | |
| | $K_F$ (mg g$^{-1}$) | 3.550 |
| | 1/n | 0.663 |
| | $R^2$ | 0.848 |
| Temkin | | |
| | $\beta$ | 23.58 |
| | $\alpha$ | 4.874 |
| | b | 96.256 |
| | $R^2$ | 0.961 |
| Jovanovich | | |
| | $K_J$ (Lg$^{-1}$) | 0.009 |
| | $q_{max}$ (mg g$^{-1}$) | 23.196 |
| | $R^2$ | 0.525 |
| Hurkins-Jura | | |
| | $A_H$ (g$^2$ L$^{-1}$) | 0.5 |
| | $B_H$ (mg$^2$ L$^{-1}$) | 2 |
| | $R^2$ | 0.588 |

### 3.2.3. Temkin Isothermal Model

This isotherm correlates surface coverage with adsorption heat. This isotherm has the following linear form [32], applied widely for elucidating adsorption data.

$$q_e = \beta \ln \alpha + \beta \ln C_e \tag{5}$$

The R is an ideal gas constant having a value of 8.314 J/mol.K, T is the absolute temperature (K), and b is a constant that can be linked to the heat of adsorption. The graph of $q_e$ (mg g$^{-1}$) vs. ln Ce (mg L$^{-1}$) is shown in Figure 2D. From the figure, the slope $\beta$ and intercept $\beta \ln \alpha$ values were determined as presented in Table 2.

### 3.2.4. Jovanovic Isotherm

This isotherm explains the mechanical connections between the adsorbent and the adsorbate. This isotherm in the linearized form [33,34] can be given as follows.

$$\ln q_e = \ln q_{max} - K_J C_e \tag{6}$$

The $q_e$ indicates the quantity of Fe(II) adsorbed per unit mass of CMPVL (mg g$^{-1}$), while the $q_{max}$ indicates the greatest removal of iron(II) estimated from the graph's intercept and $C_e$ equilibrium concentration (mg L$^{-1}$). The plot of $\ln q_e$ vs. $C_e$ is presented in Figure 2E. The slope and intercept of the curve were used to compute the values of $K_J$ and $q_{max}$, as shown in Table 2.

### 3.2.5. Harkins–Jura Model

The Harkin–Jura model describes multilayer adsorption on adsorbent surfaces with varied pore allocation. The linear form of this model can be given as follows [35]:

$$\frac{1}{q_{e^2}} = \frac{B_H}{A_H} - \frac{1}{A_H} \log C_e \tag{7}$$

The $A_H$ and $B_H$ are called Harkins–Jura constant and their values can be calculated from the $1/q_{e^2}$ vs. $\log C_{e\,plot}$ plot as shown in Figure 2F and are given in Table 2.

### 3.3. Effect of Contact Time and Kinetic Study

Figure 3A illustrates the effect of contact time on the biosorption of iron(II) on CMPVL. The adsorption of Fe(II) on CMPVL increases with time up to 10 min which is due to availability of vacant pores in greater quantity for adsorbate. Then after the fast stage, the biosorption process becomes slower indicating a steady rate of adsorption that finally led to the saturation of adsorbent sites. Equilibrium has been achieved within 60 min. Pseudo-first- and second-order, power function, intraparticle, and Natarajan models were applied to the kinetics experimental data to estimate the kinetic parameters of Fe(II) biosorption on CMPNL.

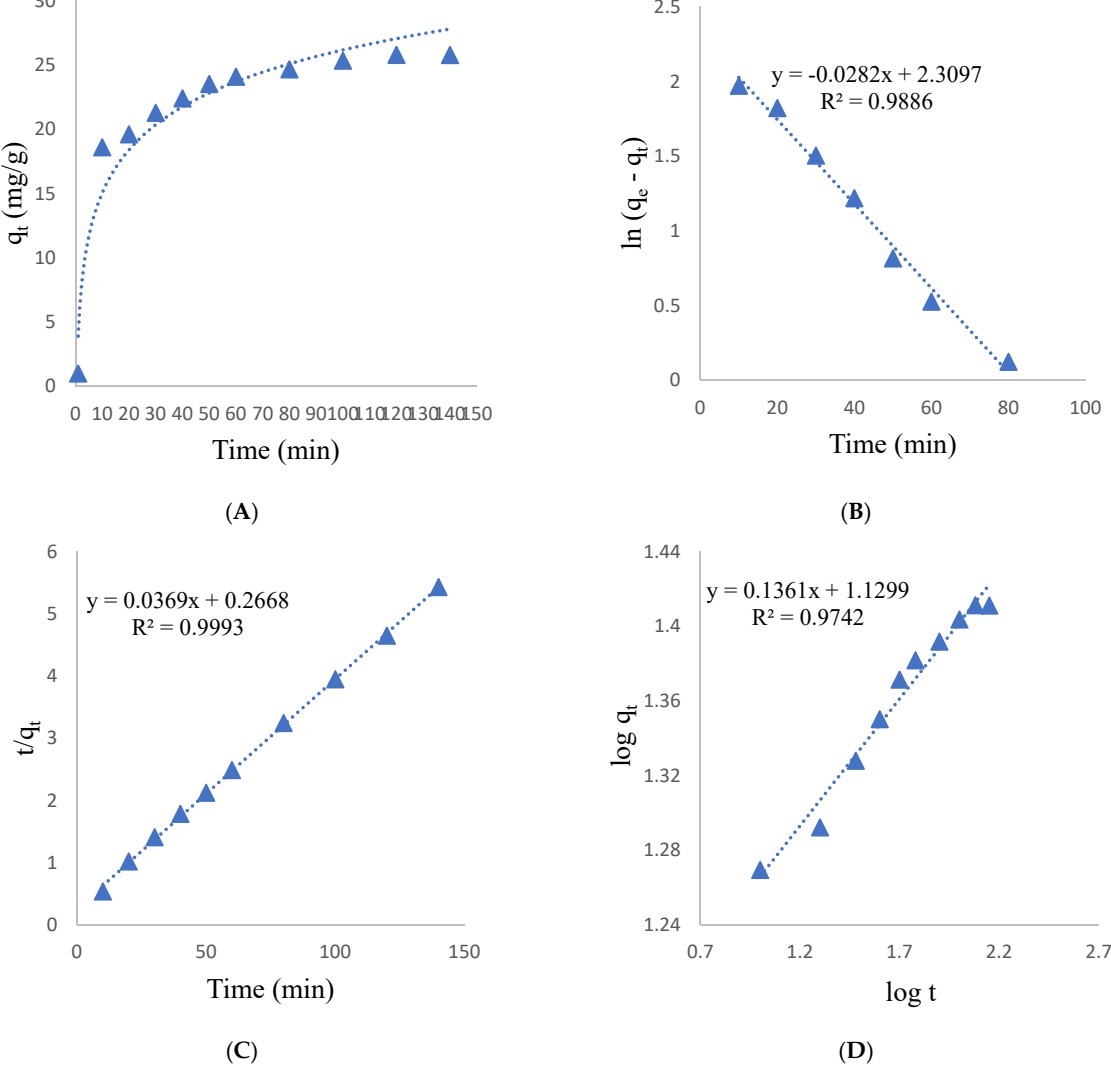

**Figure 3.** *Cont.*

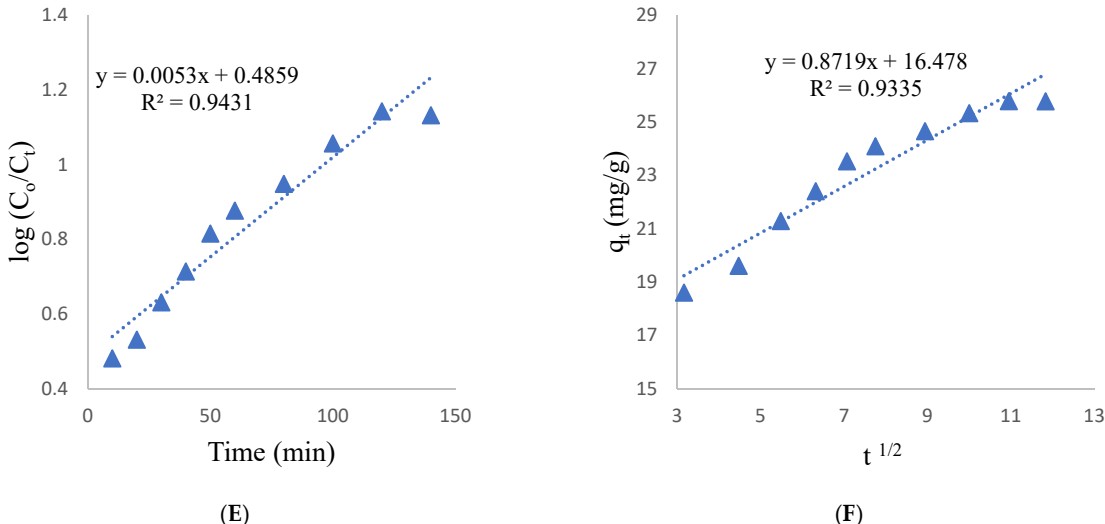

**Figure 3.** Kinetics of the iron biosorption on prepared biosorbent: (**A**) Effect of contact time (**B**) Pseudo-first-order kinetic model (**C**) Pseudo-second-order kinetic model (**D**) Power function kinetic model (**E**) Intra particular diffusion model (**F**) Natarajan and Khalaf kinetic model.

### 3.3.1. Pseudo-First-Order Kinetic Model

The mathematical form of this model is given below [36].

$$\ln(q_e - q_t) = \ln q_e - K_1 t \tag{8}$$

The $q_e$ (mg g$^{-1}$) in this equation is the capacity of Fe(II) adsorbed at equilibrium, $q_t$ is the quantity of Fe(II) adsorbed at any given time t, and $K_1$ (min$^{-1}$) is the first-order equation's rate constant. The slope and intercept of the $\ln(q_e - q_t)$ vs. t plot, as shown in Figure 3B, were used to compute the values of $K_1$ and $q_e$, which are listed in Table 3.

**Table 3.** Kinetic parameters for Fe(II) biosorption on CMPVL.

| Kinetic Model | Parameters | Values |
|---|---|---|
| Pseudo-first-order | | |
| | $K_1$ (min$^{-1}$) | $-0.028$ |
| | $q_e$ (mg g$^{-1}$) | 10.064 |
| | $R^2$ | 0.988 |
| Pseudo-second-order | | |
| | $K_2$ (min$^{-1}$) | 0.00487 |
| | $q_e$ (mg g$^{-1}$) | 27.77 |
| | $R^2$ | 0.999 |
| Power function | | |
| | $\alpha$ | 13.4586 |
| | b | 0.136 |
| | $R^2$ | 0.974 |
| Natarajan and khalaf | | |
| | $K_N$ (min$^{-1}$) | $1.15 \times 10^{-2}$ |
| | $R^2$ | 0.943 |
| Intra particular diffusion | | |
| | $K_{diff}$ (mg/g min$^{1/2}$) | $5 \times 10^{-3}$ |
| | C | 0.485 |
| | $R^2$ | 0.943 |

### 3.3.2. Pseudo-Second-Order Kinetic Model

The linear version of the pseudo-second-order kinetic equation can be given as follows [37].

$$\frac{t}{q_t} = \frac{1}{K_2 q_{e^2}} + \frac{t}{q_e} \tag{9}$$

$K_2$ (g mg$^{-1}$ min) is the pseudo-second-order rate constant, $q_e$ (mg g$^{-1}$) is the amount of Fe(II) biosorbed at equilibrium, and $q_t$ is the extent of Fe(II) biosorbed at any time t. The intercept and slope of the $t/q_t$ vs. t plot as presented in Figure 3C were used to estimate the values of $q_e$ and $K_2$. The values of constant parameters estimated are listed in Table 3.

### 3.3.3. Power Function Kinetic Model

The power function kinetic model in its linearized format [38] can be given as:

$$\log q_e = \log a + b \log t \tag{10}$$

The initial rate of adsorption is indicated by constant a, while the reaction rate can be described by constant b in Equation (10). The intercept and slope of the log $q_t$ vs. log t plot, as presented in Figure 3D, can be used to calculate both of these constant's values (Table 3).

### 3.3.4. Intraparticle Diffusion Model

The linear form of the intraparticle diffusion model is presented as follows [39,40]:

$$q_e = K_{diff} t^{\frac{1}{2}} + C \tag{11}$$

In Equation (11): $q_t$ is the amount of metal adsorbed per unit mass of CMPVL at any time t, $K_{diff}$ (mg g$^{-1}$ min$^{1/2}$) is the rate constant of the intraparticle diffusion model. The constant C (mg g$^{-1}$) is related to the boundary layer thickness and can be determined from the intercept of the $q_t$ vs. $t^{1/2}$ graph. Their values are listed in Table 3. Figure 3E shows the intraparticle diffusion curve that has been obtained by plotting $q_t$ vs. $t^{1/2}$.

### 3.3.5. Natarajan Khalaf Kinetic Model

The Natarajan Khalaf kinetic equation [41] in its linear form can be given as;

$$\log\left(\frac{C_0}{C_t}\right) = \frac{K_N}{2.303} t \tag{12}$$

In Equation (12), $C_0$ and $C_t$ are the initial and at time t concentrations of iron. The rate constant (min$^{-1}$) values are given in Table 3 which have been obtained from the slope of the log($C_0/C_t$) versus the t plot as presented in Figure 3F. According to the $R^2$ values of the models, a pseudo-second-order model with a high $R^2$ (0.999) value provided the finest match to the kinetics data.

### 3.4. pH Effect

The impact of pH on Iron(II) biosorption was studied in the pH range of 2–9 as presented in Figure 4A. The percent uptake of iron rises till pH 6, in other words, at pH 6 the maximal removal of iron has been recorded (24.367 mg g$^{-1}$). At lower pH, the adsorbent surface has been protonated by acid, and iron being a positively charged ion was thus taken up by adsorbent in lower quantity whereas at near neutral pH the interaction was favorable and thus the high adsorption capacity was recorded. The metal uptake capability was reduced at pH 7 to 9. As a result, the biosorption of iron(II) rose considerably. Iron interacts with OH$^{-1}$ at a higher pH, generating hydroxide forms that have limited the metal uptake capacity.

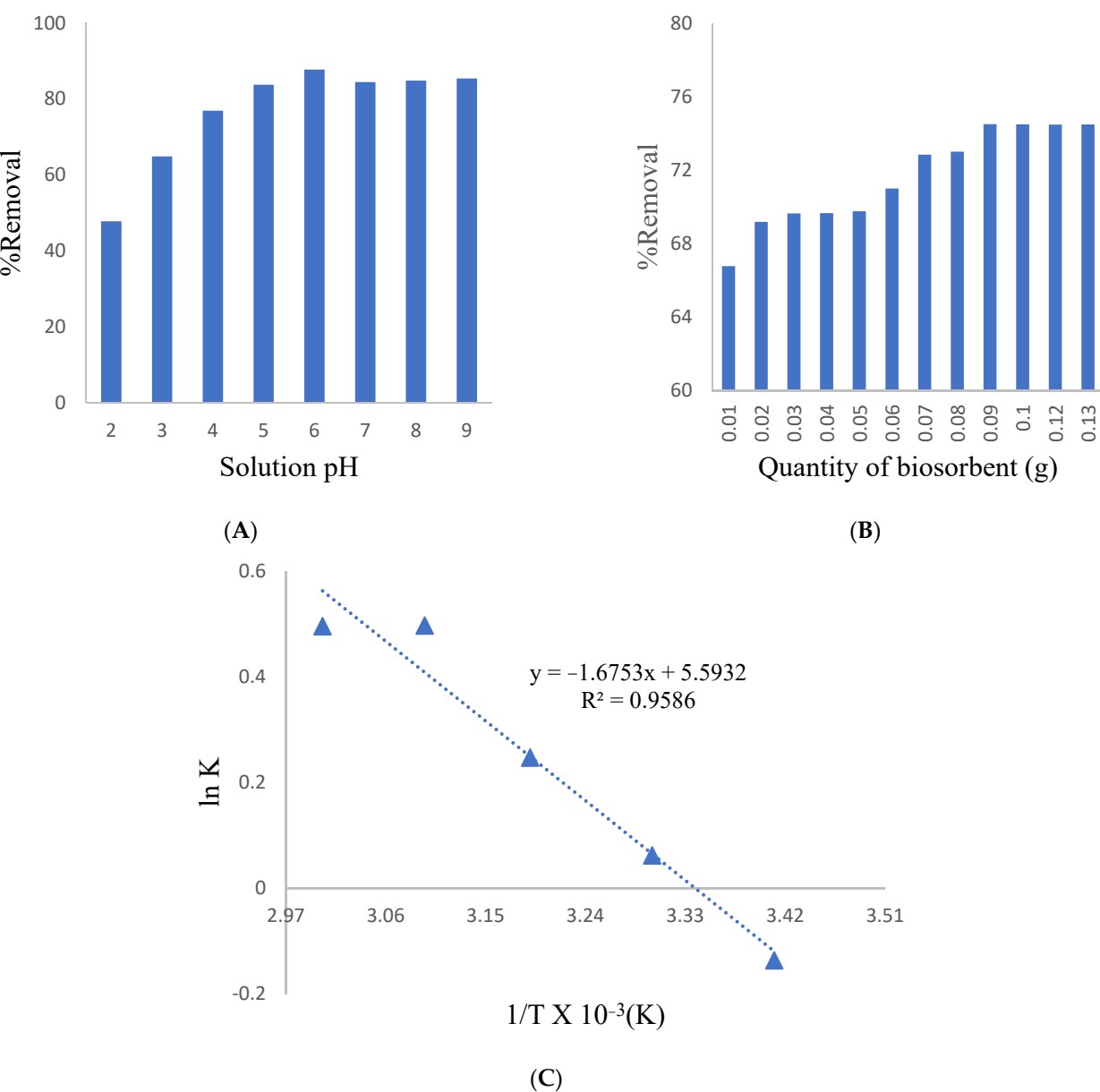

**Figure 4.** (**A**) Effect of pH (**B**) Effect of mass of biosorbent (**C**) Van't Hoff plot.

### 3.5. Effect of Mass of Biosorbent

Figure 4B depicts the influence of the mass of biosorbent dose on Fe(II) biosorption. The biosorbent dose was increased from 0.01 to 0.13 g, and the effectiveness of iron(II) elimination was measured. The optimum elimination of Fe(II) 74.5% was recorded at a 0.09 g biosorbent dose. Therefore, an optimum biosorbent dose of 0.09 g was used in all the subsequent tests. The reduced percentage removal at larger doses could be due to biomass aggregation, which limits the bio sorbent's active surface area.

### 3.6. Thermodynamic Parameters

Adsorption tests were conducted at temperatures of 293, 303, 313, 323, and 333 K to determine the thermodynamic parameters. The $\Delta H°$ and $\Delta S°$ were obtained by employing the Van't Hoff plot [42]. The following equation is used to create the plot:

$$\ln K = \frac{\Delta S^0}{R} - \frac{\Delta H^0}{RT} \tag{13}$$

where $K = q_e/C_e$ denotes adsorption ability, and $q_e$ (mg g$^{-1}$) is the amount of Fe(II) adsorbed at equilibrium. The $C_e$ stands for Fe(II) equilibrium concentration, *R* for general gas constant and *T* taken in Kelvin scale represents temperature. The $\Delta H^\circ$ and $\Delta S^\circ$ values for Fe(II) were computed using the lnK vs. 1/T plot shown in Figure 4C and are listed in Table 4. The negative $\Delta H^\circ$ and positive $\Delta S^\circ$ values reveal the exothermic and spontaneous aspects of the process. The $\Delta G^\circ$ was determined by applying the formula given below [43].

$$\Delta G^\circ = \Delta H^\circ - T\Delta S^\circ \tag{14}$$

**Table 4.** Thermodynamic parameters for Fe(II) biosorption on CMPVL.

| Biosorbent | CMPVL |
|---|---|
| $\Delta H^\circ$ (J/mol K) | $-13.92$ |
| $\Delta S^\circ$ (J/mol K) | 46.5 |
| $\Delta G^\circ$ (kJ/mol) | |
| 293 K | $-13.638$ |
| 303 K | $-14.10$ |
| 313 K | $-14.568$ |
| 323 K | $-15.033$ |
| 333 K | $-15.49$ |

The negative $\Delta G^\circ$ values indicated a favorable and spontaneous process. The value of $\Delta G^\circ$ increases with respect to temperature indicating that the sorption process is a feasible one.

### 3.7. Regeneration of the Biosorbent

The prepared biosorbent was regenerated and reused for five cycles where the drop in iron removal efficiency was 24%, which is a good sign that the adsorbent could be effectively used repeatedly in long runs (Figure 5).

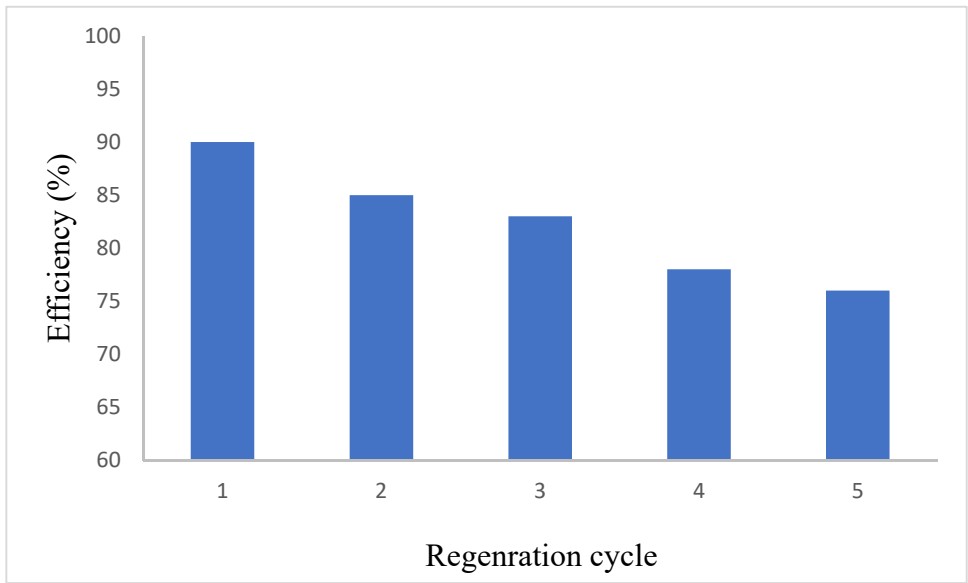

**Figure 5.** Decrease in percent efficiency of iron removal for consecutive 5 cycles.

*3.8. Comparison of Adsorption Capacities of Present Adsorbents with Those Reported in Literature*

A comparison of the adsorption capacities of the reported adsorbents with the present one has been given in Table 5. The adsorption capacities of the present adsorbent are comparatively higher than the reported ones.

**Table 5.** Comparison of present adsorbent capacity with those reported in literature.

| S. NO | Biosorbent | $q_{max}$ (mg g$^{-1}$) | References |
|---|---|---|---|
| 1 | This research work | 100 | |
| 2 | *Melia azedarach* leaves | 38.46 | [44] |
| 3 | *Waste Tea* leaves | 79.53 | [45] |
| 4 | *Typha australis* leaves | 0.84 | [46] |
| 5 | *Olive* leaves powder | 31.45 | [47] |
| 6 | *Launea procumbens* leaves | 17 | [48] |

**4. Conclusions**

Adsorption is the most environment-friendly method of removing pollutants from an aqueous solution even at ppb levels and a versatile method that has been further correlated with greener phytoremediation in this study. Among the growing plants near the drainage lines of the selected locality where steel mills are installed, *Pteris vittata* was found as the best phytoremediator of iron. Due to the slow growth process, the leaves of this plant were converted into biosorbent to achieve rapid removal of iron from the effluents. The chemically modified leaves were subjected to characterization using FTIR, surface area analyzer, SEM, TGA, and EDX techniques. The influence of pH, biosorbent dose, contact time, initial metal concentration, and temperature on the optimal uptake of Iron(II) from solution was also investigated. The maximum elimination of iron occurred at a pH of 6 and a temperature of 323 K. The Langmuir isotherm and pseudo-second-order kinetic model matched the equilibrium and kinetics data with high R$^2$ values of 0.977 and 0.999, respectively. According to the thermodynamic characteristics, iron(II) biosorption on CMPVL was favorable, exothermic, and spontaneous. From the results, it could be concluded that CMPVL, as a low-cost and readily available biosorbent with high phytoremediation capacity, could be effectively utilized to remove iron(II) from aqueous media. The prepared biosorbent employed in this study needs some improvement to increase its biosorption capacity further and also needs to be examined for the biosorption of other metals.

**Author Contributions:** Conceptualization, S.M.S. and M.Z.; methodology, Q.K., M.W. and M.Z.; N.G. and M.Z.; validation, QK.; formal analysis, F.A.K. and I.Z.; investigation, I.Z.; resources, M.Z.; writing original draft preparation: Q.K. and M.Z.; project administration: M.Z. All authors have read and agreed to the published version of the manuscript.

**Funding:** The authors declare no funding resources.

**Institutional Review Board Statement:** Not applicable.

**Informed Consent Statement:** Not applicable.

**Data Availability Statement:** The date presented in this study are available on request from the corresponding author.

**Acknowledgments:** The authors are thankful to University of Malakand for providing the research facilities.

**Conflicts of Interest:** The authors declare no conflict of interest.

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
