# Peer review of "Removal of Iron(II) from Effluents of Steel Mills Using Chemically Modified Pteris vittata Plant Leaves Utilizing the Idea of Phytoremediation"

_water, doi:10.3390/w14132004_

Round 1
Reviewer 1 Report
In this manuscript, the authors describe the removal of iron from effluents using chemically modified Pteris vittata plant leaves. The study is not that novel and lacks proper justification. Also, the results are not properly discussed. My major comments include:
1- The Author should enhance the novelty and importance of this work in the introduction section.
2- The similar studies, particularly those based on plant leaves adsorbents should be cited and discussed to highlight the importance and novelty of the developed system.
3- What is the reason to study the adsorption of iron? Is this metal more abundant in wastewater as compared to others? It is more toxic?
4- Regarding to the FTIR measurements, it should be important to mention the scans used and also the resolution. Did the authors performed the FTIR measurements using the ATR mode or using KBr pellets?
5- In specific surface area analysis, which pressure and temperature were used for degassing procedure? Which P/P0 values were used in the calculation of specific surface areas?
6- Lines 130-131: There are several typos in the manuscript.
7- The authors should make comparison with literature for the pollutant used.
8- Which type of water was used to prepare the solutions with de dyes: ultra-pure water, distilled water or deionized water? This detail is important for the readers.
9- Please do not use linearization of the equations. Nowadays, most computer programs can perform non-linear regression and should be used in preference of linearization to determine adsorption parameters.
10- Conclusions need to be improved by specifying the discussed important points within this work. In the conclusions, the authors should also provide an outlook of the challenges and potential future directions.
Other comments:
- The biggest concern with the approach presented here is the potential for sorption of unintended species in the water. The authors show that the sorbent can capture the target pollutant, but this study does not evaluate how many interfering species are also captured, which will limit the practical effectiveness. This topic should be addressed explicitly in the discussion.
- Could you comment on whether there is aging of the samples during time? This is absolutely important for practical applications.
- In order to make the adsorption process more feasible, the adsorbent is usually regenerated. Data regarding the recycling performance of the material should be added in the manuscript. Moreover, please provide the XRD and SEM, results of the adsorbent after the five times of regeneration.
- The major drawback of this paper is followed by several questions: Can this work be feasible to be done in industrial scale, and can it be scaled up? What is the novelty of this work in context of related composites used for the removal of other similar effluents compositions?
Author Response
Reviewer 1
In this manuscript, the authors describe the removal of iron from effluents using chemically modified Pteris vittata plant leaves. The study is not that novel and lacks proper justification. Also, the results are not properly discussed. My major comments include:
1- The Author should enhance the novelty and importance of this work in the introduction section.
- Worthy reviewer, the novelty statement was revised accordingly in the last paragraph of introduction section. Hopefully it will be ok now.
2-The similar studies, particularly those based on plant leaves adsorbents should be cited and discussed to highlight the importance and novelty of the developed system.
- Ans: Worthy reviewer as per your valuable suggestion, for comparison Table 5 has been inserted in the revised paper,
3- What is the reason to study the adsorption of iron? Is this metal more abundant in wastewater as compared to others? It is more toxic?
- Ans: Dargai is a free trade zone where many steel mills operates, spewing effluent into the nearby water, affecting the quality of the water and soil. The presence of iron above 0.1 mg/L damages fish gills. The concentration of iron in steel mill waste water samples was maximum according to the results of atomic absorption spectrophotometer. So iron was abundant which in that case may be toxic as compare to other metals.
4- Regarding to the FTIR measurements, it should be important to mention the scans used and also the resolution. Did the authors performed the FTIR measurements using the ATR mode or using KBr pellets?
- Ans: worthy reviewer, following detail were added accordingly.
- FTIR spectrometer Perkin Elmer were used. 0.003 g of biomass and 0.3 g of KBR was regularly assorted and then forced at 3 to 7 bar pressure to make pellet for FTIR investigation. The scanning range was 400-4000 /cm.
5- In specific surface area analysis, which pressure and temperature were used for degassing procedure? Which P/P0 values were used in the calculation of specific surface areas?
- Worthy reviewer, sorry for oversighting the error actually we have used the pycnometeric method rather than BET or BJH plots where such parameters are not involved.
6- Lines 130-131: There are several typos in the manuscript.
- Ans: The subscripts and other typos were corrected in the revised manuscript
7- The authors should make comparison with literature for the pollutant used.
- Ans: The comparison with literature is shown in the form of Table 5.
8- Which type of water was used to prepare the solutions with de dyes: ultra-pure water, distilled water or deionized water? This detail is important for the readers.
- Ans; Distilled water was used for solution preparation and washing and mentioned in 2.2 and 2.4 section.
9- Please do not use linearization of the equations. Nowadays, most computer programs can perform non-linear regression and should be used in preference of linearization to determine adsorption parameters.
- Worthy reviewer, we are not so expert in the field to apply them at this stage. However, your valuable suggestion will be endorsed in our future studies.
10- Conclusions need to be improved by specifying the discussed important points within this work. In the conclusions, the authors should also provide an outlook of the challenges and potential future directions.
- Ans: The conclusion was improved accordingly with future direction and challenges.
Other comments:
- The biggest concern with the approach presented here is the potential for sorption of unintended species in the water. The authors show that the sorbent can capture the target pollutant, but this study does not evaluate how many interfering species are also captured, which will limit the practical effectiveness. This topic should be addressed explicitly in the discussion.
- Worthy reviewer, the phytoremediation properties were first evaluated, then best phytoremediator of iron, leaves were converted into adsorbent. In the effluents concentration of iron were determined which was high. The rest of experiments have been performed on experimental water samples not real water sample and in such water samples no interfering ions are encountered. The discussion was revised accordingly.
- Could you comment on whether there is aging of the samples during time? This is absolutely important for practical applications.
- Worthy reviewer, such consideration has not been studied. In future study such parameters will be monitored. Thanks for the suggestion.
- In order to make the adsorption process more feasible, the adsorbent is usually regenerated. Data regarding the recycling performance of the material should be added in the manuscript. Moreover, please provide the XRD and SEM, results of the adsorbent after the five times of regeneration.
- Worthy reviewer, as I mentioned before we are performing the sophisticated analysis through payment as we do not have XRD or SEM facilities in our lab. The regeneration studies has been accordingly added. Figure has been added accordingly
- The major drawback of this paper is followed by several questions: Can this work be feasible to be done in industrial scale, and can it be scaled up? What is the novelty of this work in context of related composites used for the removal of other similar effluents compositions?
- Worthy reviewer, the novelty statement was accordingly provided in last part of introduction section. As far the feasibility is concerned this plant grow abundantly there near steel mills and also have high natural tendency to remediate iron. However, being researcher, we can only perform pilot scale studies not on industrial level that we have done here. These are initial reports we will extend it other metals as well.
Reviewer 2 Report
In the publication submitted for review the idea of phytoremediation was combined with the wide spread applications of adsorption in fabricating competent adsorbent for the elimination of iron from industrial effluents of steel mills. As per autors unpublished results, Pteris vittata was found to remediate iron ions effectively, thus its leaves were modified chemically to use it as efficient adsorbent for iron present in the effluents. This is a very interesting study that was carried out with different Fe concentrations. Other aspects like the pH of the solution (6.0), the volume of the solution (50 mL), the contact period (120 minutes), and the mass of the biosorbent (0.09 g) remained constant. The influence of pH, biosorbent dose, contact time, initial metal concentration, and temperature on the optimal uptake of Iron(II) from aqueous solution were investigated. Maximum elimination of iron occurred at a pH of 6.
· However, there is no information about Pteris vittata in the introduction. Is it a common plant where it is to be used?
· It is interesting to compare the presented studies with the results obtained for other pH parameters and contact times. It is very valuable.
However, the article needs some minor corrections:
· Please correct the markings in the drawings. Use either lowercase or uppercase letters to match the description.
· Please also organize and standardize axis descriptions in the figures (units).
· Also consider whether Figure 1 (i) can be improved. It is not compatible with the rest.
I like that the conclusions of the research have been presented in a concise form.
Moreover, the publication is a valuable source of information, especially for industry wastewater treatment l and the publication is the basis for further research.
Thank you for considering my opinion. I encourage authors to keep on working to improve the manuscript.
Author Response
Reviewer 2
In the publication submitted for review the idea of phytoremediation was combined with the wide spread applications of adsorption in fabricating competent adsorbent for the elimination of iron from industrial effluents of steel mills. As per autors unpublished results, Pteris vittata was found to remediate iron ions effectively, thus its leaves were modified chemically to use it as efficient adsorbent for iron present in the effluents. This is a very interesting study that was carried out with different Fe concentrations. Other aspects like the pH of the solution (6.0), the volume of the solution (50 mL), the contact period (120 minutes), and the mass of the biosorbent (0.09 g) remained constant. The influence of pH, biosorbent dose, contact time, initial metal concentration, and temperature on the optimal uptake of Iron(II) from aqueous solution were investigated. Maximum elimination of iron occurred at a pH of 6.
- However, there is no information about Pteris vittatain the introduction. Is it a common plant where it is to be used?
- Thank you worthy reviewer, these information were accordingly provided. However, being adsorption study its phytochemistry or family background or medicinal uses has not been provided.
- It is interesting to compare the presented studies with the results obtained for other pH parameters and contact times. It is very valuable.
- Thank you worthy reviewer, for the encouraging remarks.
However, the article needs some minor corrections:
- Please correct the markings in the drawings. Use either lowercase or uppercase letters to match the description.
- Thank you worthy reviewer, such discrepancies were accordingly removed.
- Please also organize and standardize axis descriptions in the figures (units).
- Thank you worthy reviewer, they were corrected accordingly
- Also consider whether Figure 1 (i) can be improved. It is not compatible with the rest.
- Figure quality was accordingly improved. However, most of them are scan images that is why seem different from others.
I like that the conclusions of the research have been presented in a concise form.
- Thank you worthy reviewer, for the encouraging remarks.
Moreover, the publication is a valuable source of information, especially for industry wastewater treatment l and the publication is the basis for further research.
- Thank you worthy reviewer, for the encouraging remarks.
Thank you for considering my opinion. I encourage authors to keep on working to improve the manuscript.
- Thank you worthy reviewer, for the encouraging remarks.
Reviewer 3 Report
Dear authors,
Please see the attached file where you can find my comments.
Kind regards,

Author Response
Reviewer 3
This study aimed to investigate the removal of Iron (II) from effluents of steel mills using chemically modified Pteris vittata plant leaves utilizing the idea of phytoremediation. The manuscript is recommended for publication in journal with minor revision. Some issues must be take into consideration:
- English language and style are fine/minor spell check required
- Thank you, worthy reviewer, for the positive input. Changes where required were made accordingly.
-In the Abstract section it is mentioned that: ‘’Initially, the effluents were analyzed for heavy metal concentrations, then leaves of a plant (Pteris vittata) with better phytoremediation capability was used as efficient adsorbent’’. Therefore, please add in the manuscript some results in order to sustain the choice for Pteris vittata as adsorbent for Fe(II)
- Thank you worthy reviewer, the abstract was revised accordingly.
-Please, add the novelty statement
- The statement has been made in last paragraph of the introduction section accordingly.
-Please, add a comparison with the literature
- Thank you worthy reviewer, the comparison has been presented in the form of Table 5.
-Please, add the adsorbent’s reusability results taking into consideration that at L102 it is mentioned that: ‘’ As a result, the adsorbent's reusability was also assessed.’’
- Thank you worthy reviewer, the required details were incorporated accordingly.
-The References section needs to be revised. The references notes differ (The journal recommends preparing the references section with a bibliography software package, such as EndNote, ReferenceManager or Zotero to avoid typing mistakes and duplicated references).
- Worthy reviewer, they were accordingly revised using end note
-Please check the numbering of the references. For example: the References 29, 30, and 31 appear for the first time in the manuscript body after Reference 42. Also, Reference 42 appears for the first time in the manuscript after References 43, 44.
- The references sequence in text and list were accordingly revised.
-The manuscript must be revised: some spaces should be added. For example:
-L28: ‘’0.01-0.13’’ replace with ‘’0.01-0.13 g’’
- Replaced accordingly
- ‘’0.09g’’ replace with ‘’0.09 g’’
- Replaced accordingly
-L166: ‘’32001/cm’’ replace ‘’with 3200 cm-1’’
- Replaced accordingly
-L187: ‘’8.286mg’’ replace with ‘’8.286 mg’’
- Replaced accordingly
-L192: ‘’7.873mg’’ replace with ‘’7.873 mg’’
- Replaced accordingly
- ‘’mgg-1’’ replace with ‘’mg g-1’’ in all manuscript body
- Replaced accordingly
-‘’mgL-1’’ replace with ‘’mg L-1’’ in all manuscript body
- Replaced accordingly
-Table 2: ‘’Lmg-1’’ replace with ‘’L mg-1’’; ‘’Lg-1’’ replace with ‘’L g-1’’
- Replaced accordingly
-L16: Please, verify the font
- Corrected accordingly
-L34: Please, replace ‘’Van,t’’ with ‘’Van’t’’
- Replaced accordingly
-L45-L47: In my opinion this information must be rephrase: ‘’Water being the essential commodity of life in prone to heavy metal contamination by rapid industrialization around the globe.’’
- The sentence was rephrased accordingly.
-L130: Subscript ‘’FeSO4.7H2O’’
- Corrected accordingly.
-L165-170: In my opinion 1/cm should be replace with cm-1
- Replaced accordingly
Figure 1: Fig. A and B can be presented in one figure. The same observation for Fig. H and I
- Thank you worthy reviewer, however, we are not expert of these software to combine them
-L252: Superscript 1/qe2
- Corrected accordingly.
L288: Add a point at the end of the paragraph.
- Point was added at the end of paragraph.
-L302: Remove the point between ‘’Figure 3.’’ and ‘’(f)’’
- Corrected accordingly
-L312: Subscript HNO3
- Corrected accordingly
-L325: Please correct the information: ‘’The biosorbent dose was increased from 0.01 to 0.14 g, and
the effectiveness of iron (II) elimination was measured’’
- Corrected accordingly
-L326: Please replace ‘’74.5 percent’’ with ‘’74.5%’’
- Replaced accordingly.
-Figure 4 (a,b) : The unit for the Percentage removal must be indicated on y-axis
Ans; The %R indicated on Y axis.
- Corrected accordingly
Round 2
Reviewer 1 Report
Authors have revised the manuscript according the recommendations, and answered the questioned points. Now it looks suitable for publication.
Author Response
Thank you worthy reviewer for your positive input